# Spontaneous Sepsis in Adult Horses: From Veterinary to Human Medicine Perspectives

**DOI:** 10.3390/cells12071052

**Published:** 2023-03-30

**Authors:** Angélique Blangy-Letheule, Amandine Vergnaud, Thomas Dupas, Bertrand Rozec, Benjamin Lauzier, Aurélia A. Leroux

**Affiliations:** 1Nantes Université, CHU Nantes, CNRS, INSERM, l’institut du thorax, F-44000 Nantes, France; 2CHU Sainte-Justine Research Center, Montreal, QC H3T 1C5, Canada; 3Department of Clinical Sciences, Equine Veterinary Teaching Hospital (CISCO), Oniris, F-44300 Nantes, France

**Keywords:** sepsis, animal model, horse, biomarkers, therapeutic strategies

## Abstract

Sepsis is a life-threatening disease defined as an organ dysfunction caused by a dysregulated host response to an infection. Early diagnosis and prognosis of sepsis are necessary for specific and timely treatment. However, no predictive biomarkers or therapeutic targets are available yet, mainly due to the lack of a pertinent model. A better understanding of the pathophysiological mechanisms associated with sepsis will allow for earlier and more appropriate management. For this purpose, experimental models of sepsis have been set up to decipher the progression and pathophysiology of human sepsis but also to identify new biomarkers or therapeutic targets. These experimental models, although imperfect, have mostly been performed on a murine model. However, due to the different pathophysiology of the species, the results obtained in these studies are difficult to transpose to humans. This underlines the importance of identifying pertinent situations to improve patient care. As humans, horses have the predisposition to develop sepsis spontaneously and may be a promising model for spontaneous sepsis. This review proposes to give first an overview of the different animal species used to model human sepsis, and, secondly, to focus on adult equine sepsis as a spontaneous model of sepsis and its potential implications for human and veterinary medicine.

## 1. Introduction

Since the 2016 Sepsis-3 consensus, sepsis has been defined as an organ dysfunction caused by a dysregulated host response to an infection, putting the host’s life at risk [1]. In 2017, this condition affected 48.9 million people worldwide, resulting in 11 million deaths, or approximately 20% of worldwide mortality [2]. In intensive care units, this pathology is the leading cause of death [3]. Despite the prevention measures practiced in hospitals worldwide, the incidences of sepsis and associated mortality are constantly increasing (incidence +7.2% each year between 1993 and 2010 in France) [4,5]. This mortality could be explained by the fact that sepsis is a complex and multifactorial disease that presents great heterogeneity between patients, with no pertinent model to study this condition. Indeed, the clinical manifestations of sepsis depend on many parameters, such as the initial site of infection, the responsible microorganism, the underlying health status of the patient, or the delay before the start of diagnosis and treatment. All of these factors complicate appropriate management. Early diagnosis of sepsis and timely therapeutic interventions are essential to improve clinical outcomes and reduce mortality [4]. Since 2017, the World Health Organization has made sepsis a global priority and therapeutic emergency and has passed resolutions to improve the prevention, diagnosis, and management of this condition [6]. Due to its global impact, sepsis continues to be the subject of intense research in which the current understanding of the pathophysiology is being re-evaluated, and the relevance of various in vivo models is being questioned. This reflects the importance of experimental animal models and, in particular, the animal species chosen to be relevant to human pathology. Spontaneous sepsis is a common condition in veterinary medicine, especially in equids, because of their susceptibility to developing bacterial endotoxins and their predisposition to develop gastrointestinal diseases such as colic and diarrhea. As for humans, gastrointestinal and pulmonary diseases are the main causes of spontaneous sepsis in horses. The pathophysiology, clinical manifestations of equine sepsis, its immune consequences, and some biomarkers of sepsis, such as procalcitonin or serum amyloid A (SAA), appear to be similar between horses and humans. Investigation of a spontaneous model of sepsis could allow, unlike experimental models, to replicate both clinical complexity and medical management. However, such studies are still rare. As equine sepsis becomes better understood, clinicians, veterinarians, and scientists will be able to develop new therapeutic approaches or targets and even improve the prevention of this devastating disease in humans and horses. The purpose of this review is to discuss the potential benefits of increased knowledge of equine sepsis for human and veterinary medicine.

## 2. Human Definition of SIRS and MODS

In 1991, a consensus on definitions of sepsis sponsored by the American College of Chest Physicians and the Society of Critical Care Medicine defined the concept of a systemic inflammatory response syndrome (SIRS) to describe the pathophysiological response that results from infection, trauma, burns, or pancreatitis [7]. In humans, it consists of four clinical manifestations: (i) tachycardia (heart rate > 90 beats/min), (ii) tachypnea (respiratory rate > 20 breaths/min), (iii) fever or hypothermia (temperature > 38 or <36 °C), and (iv) altered white blood cells (white blood cell count > 12,000/mm^3^ or <4000/mm^3^), or the presence of immature neutrophils (≥10%) [7]. According to the Sepsis-1 consensus, the presence of two of these criteria is sufficient to demonstrate a systemic inflammatory response.

In 1991, an international consensus formally defined sepsis as “a systemic inflammatory response of a host to an infection” (Sepsis-1) [7]. At the time of this consensus, the term severe sepsis was defined as sepsis associated with organ dysfunction, hypoperfusion, or hypotension, which can evolve into septic shock. In 2001, a second consensus (Sepsis-2) recognized the limitations of the first definition and expanded the list of associated clinical manifestations. Sepsis-2 defines septic shock as a state of acute circulatory failure [8]. This definition remained unchanged for almost two decades. Following the improvement of knowledge of pathophysiological mechanisms, a third international consensus redefined the notion of sepsis and septic shock. Today, sepsis is defined as an organ dysfunction caused by a dysregulated host response to an infection that is life-threatening. Septic shock is characterized by persistent hypotension associated with cellular metabolic failure, coupled with hyperlactatemia reflecting significant tissue dysoxia (>2 mmol/L) [1].

Organ dysfunction, also known as multiple organ dysfunction syndrome (MODS), now central to the clinical definition of septic shock, is a dynamic process that can be reversible if detected and managed early. In human medicine, the multi-organ dysfunction score, developed in 1995 by Marshall and colleagues, was based on an observational cohort study, which was tested for validity in 692 patients. This score was introduced in the 2001 consensus to define organ dysfunction in severe sepsis [8]. In human medicine, MODS is identified by a change of at least two points in the sequential (sepsis-related) organ failure assessment (SOFA) score following an infection. This score assesses different parameters such as the respiratory, hematological, cardiovascular, hepatic, renal, and central nervous systems of patients and assigns each parameter a value between 0 (normal organ function) and 4 (most abnormal organ function) (Table 1). Since the introduction of MODS scoring systems for critically ill patients, overall MODS severity and mortality rates have decreased, which may be due to earlier recognition of organ dysfunction [1].

## 3. Preclinical Experimental Animal Models of Sepsis Benefits for Human Medicine

Animal models contribute to the understanding of human diseases by providing complex environments and validating therapeutic targets and/or biomarkers. In this context, standardization, reproducibility, and translation of animal species into human medicine must be considered to reach useful and effective conclusions.

The relevance of experimental models of sepsis is based on two crucial elements. The first is the choice of the model used to induce sepsis. In the literature, several experimental models of sepsis, such as endotoxin administration, bacterial infusion, fecal slurry, cecal ligation and puncture (CLP), and colon ascendants stent peritonitis, have been described. These models have advantages and disadvantages which have been discussed in the previous review and will not be discussed in this review [10,11]. The second crucial element is the choice of animal species. This review will focus on this last point.

Rodents’ models of sepsis are frequently used in research settings. The advantages of these models are their ease of implementation and the animals’ standardization which translates into a similar genetic background, and the same age, gender, living environment, and weight. In addition, a wider range of reagents is available for immunological studies in rodents than any other species (Table 2). These models have allowed a better understanding of the pathophysiological mechanisms and the identification of new biomarker candidates and new potential therapeutic targets. However, mice and rats are resistant to all procedures, and they do not perfectly mirror the pathophysiological mechanisms observed in humans [10]. Research on sepsis over the last 40 years has led to more than 30 clinical trials, which, despite their success in murine models, have proved ineffective and even deleterious in humans. These failures can partly be explained by the fact that gene expression shifts during SIRS are different between mice and humans [12]. In the face of numerous therapeutic failures, these animal models are increasingly being questioned [13].

Large animal models have been developed to better reproduce human sepsis. These models have a larger blood volume which allows for larger blood samples to be taken. In addition, the medical instruments used in large-animal veterinary medicine are similar to those used in human medicine [15].

Canine models of endotoxemia and bacteremia have been widely used to study cardiovascular function during sepsis. Like humans, dogs show an increase in heart rate, a decrease in blood pressure and leukocytosis, and an increase in plasma adrenaline and noradrenaline during septic shock [16]. Furthermore, in 2012, Song and colleagues reported increased interleukin 6 (IL-6) mRNA in peripheral blood mononuclear cells and increased tumor necrosis factor alpha (TNF-α) in the first few hours after induction of endotoxemia compared with healthy controls [17]. However, unlike humans, dogs are resistant to endotoxins and require a high dose of lipopolysaccharides (LPS) to induce sepsis [18,19].

Farm animals, including swine and small ruminants, are commonly used in large experimental animal models. Their large size is more representative of humans, and they are anatomically and physiologically closer to humans. Because of their close relationship, farm animals also share many pathogens with humans [20]. The main advantages and disadvantages of the different farm animal models as induced sepsis models are detailed in Table 2. The use of these farm animals as models of human disease has long been precluded by the paucity of genomic information compared to other animal models, such as mice. Advances in genomics have made it possible to obtain the porcine genome [21]. From an evolutionary point of view, the swine is the closest non-primate mammal species to humans. The anatomy and physiology of several organs, such as the eye, heart, liver, and gastroenteric system, have similarities with humans [22]. Indeed, swine, like humans and unlike most other large mammals, are omnivorous. Swine also share with humans their sensitivity to certain bacterial pathologies [20]. There is a wide variety of natural and transgenic swine lines on the market [23]. These genetic and physiological similarities between swine and humans make this species one of the most promising experimental models for human clinical applications [24]. The swine model has been used to study new therapeutic targets to improve renal, hepatic, intestinal, or cardiovascular function during sepsis [25,26]. However, current equipment and protocols make it difficult to keep swine in an intensive care unit because of complications related to mechanical ventilation, which does not translate well into clinical care performed in humans. In addition, swine are particularly susceptible to the development of impaired pulmonary function associated with pronounced acute pulmonary hypertension, which is less representative of human pathophysiology.

The ovine model of sepsis shows similarities in cardiopulmonary response to human sepsis. In contrast to the porcine model, the ovine model of sepsis has been particularly used to study lung injury associated with sepsis. In 2011, Maybauer and colleagues used this model to define the pathogenetic role of peroxynitrite and its catalyst WW-85, demonstrating an overall improvement in lung function [27]. Furthermore, like humans, sheep are more sensitive to LPS than swine models. Continuous infusion of LPS at a rate as low as 9 ng/kg per hour results in marked changes in pulmonary arterial pressure, cardiac output, and pulmonary microvascular permeability [28]. However, sheep are ruminants, and their gastrointestinal anatomy and physiology markedly differ from those of omnivorous species such as humans [14].

However, all of these induced-experimental animal models require complex medical equipment, are difficult to house due to their size, and are more difficult to study on a molecular basis due to the limited number of biological reagents. In addition, large-animal-induced models are very expensive, which makes it difficult to obtain large cohorts of animals. Finally, it should be kept in mind that the limitations regarding sex and age that exist in mouse models persist when using larger species for experimental purposes. It is becoming clear that there is an important need for a new animal model to better reproduce human sepsis. The most common criticisms of many experimental animal models of sepsis include the use of unmixed young animals, lack of sensitivity to infection, and management, which fail to reflect clinical reality.

Experimental models of sepsis have difficulty in faithfully reproducing the pathophysiology of human sepsis and its medical management issues, while this pathology is also observed in veterinary medicine, with animals developing this pathology spontaneously. These animals have hospital management similar to that used in humans. The study of spontaneous sepsis in animals could represent an interesting step in the understanding of pathophysiological mechanisms and in the discovery of new biomarkers or therapeutic targets while preserving the heterogeneity observed in humans. To our knowledge, the large animal species described above are rarely hospitalized following spontaneous sepsis. This fact can be explained by resistance to infections but also for economic and health reasons. Indeed, swine and ovine are exceptionally hospitalized when they develop spontaneous sepsis because of the cost of the treatment and because the admission of some drugs can make them improper for consumption. Among the species developing spontaneous sepsis requiring hospitalization, horses seem to be a good candidate for the study of sepsis and will be discussed further in this review.

## 4. Spontaneous Equine Sepsis, a Pertinent Veterinary Disease for Modeling Human Sepsis

Horses appear to be more prone to develop sepsis than other large animal species. Experimental data suggest that, like humans, they are much more susceptible to the effects of endotoxins [29]. This higher sensitivity of the horse to LPS has been demonstrated in studies using equine models of LPS injection or infusion in horses [30,31,32]. In the majority of these studies, a dose of 30 ng/kg of LPS was administered to observe signs of sepsis, whereas, for swine models, the doses used were 100 times higher (µg/kg) [25,33]. Furthermore, like humans, and unlike experimental models, horses live in an environment rich in microbial exposure. Similar to what is observed in humans, sepsis of abdominal origin in horses can occur in healthy animals without underlying immune pathology. A few studies have begun to focus on spontaneous equine sepsis, notably that of Anderson and colleagues, who sought to identify proteomic profile characteristics to distinguish septic from non-septic equine arthropathies [34].

The advantage of this approach is that, as for humans, horses admitted with sepsis have a different age, gender, body mass index, and co-morbidities at admission, which mimic the reality of hospital care. In addition, horses developing spontaneous sepsis are hospitalized and treated at different times after the onset of the first symptoms of sepsis, which replicates the conditions observed in human medicine. Veterinary management methods are based on human medical management methods. Horses hospitalized in intensive care receive intravenous fluid and antibiotic therapy as observed in human medicine, which is not the case in current experimental models. In addition, horses with sepsis show hyperlactatemia, severe cardiovascular changes such as tachycardia, abnormal mucosal appearance, increased capillary refill time, or myocardial dysfunction, as well as multiple hematobiochemical abnormalities observed in humans [35,36]. As for humans, cardiovascular shock secondary to sepsis generates organic dysfunctions that can lead to acute cardiac decompensation, endothelial dysfunction, and coagulation disorders, which cause excess mortality both in humans and horses [37]. In addition, as previously explained, many horses develop sepsis following gastrointestinal injuries or pneumonia, which would allow obtaining large cohorts to identify and validate candidate biomarkers.

## 5. SIRS and MODS Definition for Adult Horse

In equine veterinary practice, the terms “endotoxemia” or “endotoxemic shock” are still commonly used when signs of cardiovascular shock secondary to severe digestive damage are observed. One explanation could be the high incidence of endotoxemia in horses. Indeed, 29% of horses with colic show detectable levels of LPS in the blood [38]. However, the term “sepsis” is now generally preferred as the presence of bacterial endotoxins in the blood is rarely confirmed in clinical practice. In adult horses, no consensus has been reached; however, the veterinary scientific community has agreed to define sepsis with the same definition used in human medicine. Similarly, the management of horses with sepsis is mostly based on human practice. A few scientific studies have begun to focus on equine sepsis, confirming its negative impact on survival [39,40]. As for humans, the use of early biomarkers and therapeutic targets that are sensitive and specific to the evolution of the pathology would facilitate rapid diagnosis and, therefore, early management of septic horses, limiting organ dysfunction, particularly cardiac, and optimizing their chances of survival. However, to date, and as for humans, there are no biomarkers for the early and specific diagnosis of equine sepsis. It is, therefore, essential for research teams working on equine sepsis to generate better results, identify reliable biomarkers in the future, and propose new therapeutic strategies. In order to use the horse as a model of spontaneous sepsis, it is essential to be able to identify the pathology quickly and easily to avoid selection error. Considering the similarity of equine sepsis to humans, the majority of these studies have used the consensus definitions of SIRS and sepsis used in human medicine. In an ongoing effort to establish an accurate definition, veterinarians have developed various SIRS and MODS scores to aid in medical decision making.

### 5.1. Equine SIRS Definition

Many adult horses with colic exhibit clinical signs similar to those described for SIRS [41]. Although the criteria for SIRS have not been definitively defined in veterinary medicine, several equine studies have extrapolated the definition of SIRS from human medicine [35,42,43,44]. However, these studies were not performed using the same criteria for diagnosing horses in SIRS (Table 3). To be included in the Schwarz, Aguirre, and Epstein studies, the heart rate must be higher than 60 beats per minute (bpm), whereas it must be higher than 50 bpm in the Borde study. The latter also includes two additional parameters, which are hyperlactatemia (Lactate > 2 mmol/L) or hypotension characterized by a mean arterial pressure below 90 mmHg (Table 3).

In 2017, Roy and collaborators studied the clinical relevance of SIRS in an equine emergency population (*n* = 464). For this purpose, a prospective study of diagnostic test results, diagnosis, and outcomes was performed and then followed by a retrospective analysis of all data. They identified cutoff values to perform a score identifying horses as septic or not (Table 4). Horses presenting a score higher than 2 were considered in SIRS. This study confirmed that the SIRS criteria were associated with an increased risk of death. However, this study did not include an analysis of neutrophils, although they are included in the original definition of human SIRS. The use of a SIRS score in equine medicine may be useful as a categorization tool and could be used as a diagnostic criterion for sepsis. Although the score developed by Roy and colleagues has not been evaluated in a validation cohort, it is now widely used to distinguish horses with SIRS.

### 5.2. Equine MODS Definition

Despite advances in medical management, sepsis continues to be a leading cause of illness and death in horses [45,46,47,48]. The use of a robust SOFA-like score in equine medicine to easily identify horses with MODS without the need for further investigation would facilitate the comparison of different studies and limit selection bias. However, the MODS criteria have not been validated in horses, and the SOFA score incorporates physiological data such as mean arterial pressure (MAP) or inspired oxygen fraction, which are difficult to access during emergency clinical examinations on horses. These elements underline the need for an equine-specific scoring system that is validated for clinical use. Therefore, studies have sought to adapt this score to make it compatible with equine use.

In 2016, McConachie and colleagues conducted the first study to establish individual organ dysfunction criteria to reflect a range of severity of MODS development for use in adult horses with acute surgical gastrointestinal disease. For this purpose, a score based on a literature search was developed. This score is based on the results of examinations performed at admission, including a thorough clinical examination, a complete hematobiochemical analysis, transrectal palpation, nasogastric probing, abdominal ultrasound, and rapid echocardiography. This score was then retrospectively evaluated in horses with colic that required exploratory laparotomy. This study showed that a score higher than 8 on day 1 or day 2 post-surgery is associated with increased mortality 6 months after surgery [36]. Although this score has not been validated in horses that have not undergone gastrointestinal surgery, it has been adapted in several clinics to identify MODS in horses admitted for colic and suspected of sepsis. Studies evaluating the MODS in horses with sepsis that have not had surgery could be set up to validate this score. In addition, the study of easily accessible parameters in horses that have not been monitored during surgery could complement this score in order to expand its use.

SIRS and MODS scores have been developed to assist in the diagnosis of horses with sepsis. These studies have provided preliminary information regarding incidence, severity, treatment efficacy, and survival in horses. In contrast to the large number of studies that have examined the impact of sepsis on survival in hospitalized foals, studies of sepsis-induced mortality in adult horses are rare. The use of these definitions allows for accurate and applicable comparisons between populations and studies, advancing knowledge of the pathophysiologic mechanisms of sepsis in the adult horse and improving the identification of biomarkers or therapeutic targets.

## 6. Main Biomarkers of Equine Sepsis: A Proof of Concept for Human Medicine

As for humans, the greatest challenge of an equine patient with sepsis is early recognition and treatment. Thus, early diagnosis of this condition can lead to rapid and effective treatment and a better prognosis. The National Institute of Health’s Biomarker Definitions Task Force has defined an ideal biomarker as “a more rapid diagnostic tool, capable of distinguishing between a normal or pathological process, and providing information about the response to pharmacologic or otherwise therapeutic intervention.” [49]. The use of biomarkers could allow diagnosis, risk stratification, monitoring of clinical course, and therapeutic management, which would complement the information provided by the SIRS and MODS scores.

### 6.1. Main Biomarkers Currently Used to Diagnose Sepsis

Circulating protein biomarkers have played an important role in clinical decision making for infectious diseases, particularly in the assessment of disease severity and treatment efficacy. In recent years, laboratory biomarkers have become important in diagnosing sepsis, guiding antimicrobial therapy, and assessing response to therapy in human medicine. Few protein biomarkers have been studied during sepsis in adult horses, the main ones being common with biomarkers used in human medicine and having the same drawbacks of sensitivity and specificity (Table 5) and the same variation (Table 6).

#### 6.1.1. Fibrinogen

Fibrinogen, the most commonly assessed acute phase reactive protein in horses, is known to increase with surgical trauma and inflammatory/infectious conditions [54]. In inflammatory conditions, fibrinogen is involved in tissue repair and induces an intracellular signaling cascade that upregulates cell phagocytosis, degranulation, and cytotoxicity [55]. Given the central role of this precursor for fibrin generation, high levels of fibrinogen may be partly responsible for the coagulopathy and endothelial damage observed in inflammatory states. [56].

In horses with inflammatory gastrointestinal diseases such as enterocolitis and peritonitis, fibrinogen concentrations are higher than those observed in healthy horses [57,58]. In addition, increased concentrations of fibrinogen were observed in the peritoneal fluid of horses with colic [59]. It is a non-specific indicator of inflammation that is often increased in the presence of SIRS [60]. Although fibrinogen is considered an acute phase-reactant protein, values increase 24 h after induction of inflammation and might not peak for 2–3 days [55]. Furthermore, it is present at high concentrations in the blood of healthy horses, and concentrations only increase about 2–4 fold in response to an inflammatory stimulus [61,62]. These elements do not allow the detection or progression of the disease. Finally, the involvement of fibrinogen in the coagulation cascade complicates the interpretation of fibrinogen measurements. Levels may decrease due to consumption during diseases characterized by disseminated intravascular coagulation or changes in vascular permeability and thus mask increases induced by inflammatory processes [63]. In addition, chronic diseases such as renal failure are susceptible to increased fibrinogen levels [64]. Although fibrinogen measurements are relatively easy and inexpensive, the use of more sensitive biomarkers or a panel of different biomarkers may allow better discrimination between different conditions.

#### 6.1.2. D-Dimer

Pro-inflammatory cytokines, released during sepsis, activate the coagulation cascade, which promotes fibrin degradation resulting in an increase in D-dimer [65]. In human medicine, this protein is used to monitor septic patients, particularly to diagnose disseminated intravascular coagulation. It has been demonstrated in humans with sepsis that this protein is a good predictor of mortality [66]. D-dimer is a simple, inexpensive, commonly available test that reflects closely the activation of the coagulation system and, in this way, potentially also the severity of the host response. In veterinary medicine, D-dimer has been identified as a useful tool for the detection of the thrombotic state and hyper-fibrinolytic states in horses with colic [67,68,69]. Furthermore, it has been shown that horses with colic show an increase in peritoneal D-dimer concentration compared to healthy horses [70] and that this protein could be used as a diagnostic and prognostic marker for horses with severe forms of colic [71]. However, to date, D-dimer is rarely used as a biomarker of sepsis in the adult horse.

#### 6.1.3. Serum Amyloid A

Serum amyloid A (SAA) appears to play an important role in the promotion of acute inflammation. Indeed, studies have shown that SAA is involved in important immune-associated activities by inducing the production of inflammatory mediators, including TNF-α, nitric oxide (NO), and interleukin-6 (IL-6) by immune cells, such as mononuclear macrophages [72,73]. SAA has been studied in various clinical and experimental settings as an alternative to fibrinogen. It is one of the major acute-phase proteins in horses, which has been shown to be a differentiating factor for septic and non-septic inflammatory conditions. In horses with colic, SAA concentration was significantly higher in non-survivors compared to survivors and in horses with inflammatory lesions compared to healthy horses [74]. Several studies have shown an increase in SAA concentrations 12 h after surgery, with values peaking and then rapidly decreasing compared to fibrinogen, demonstrating a more progressive and persistent response [75,76]. SAA has several advantages over fibrinogen as a biomarker, including a more rapid and pronounced increase in response to inflammatory disease and greater sensitivity when used to indicate the presence of systemic inflammation in equine gastrointestinal disease [77]. Indeed, low or undetectable levels of SAA in healthy horses facilitate the interpretation of slightly elevated concentrations, and the rapid and exponential increase in plasma levels after an inflammatory stimulus allows close monitoring of disease progression [62,78]. Because of the short half-life, plasma SAA levels decline with successful treatment and resolution of the disease, whereas fibrinogen levels remain elevated for a long period after the resolution of the disease. Sequential measurements of SAA are, therefore, potentially a useful aid to patient management in the equine clinic, including the evaluation of treatment strategy [79]. For these reasons, SAA has become a popular biomarker of inflammation in horses. Although SAA is a more sensitive indicator of inflammation than fibrinogen, it is not specific to inflammation [80].

#### 6.1.4. C-Reactive Protein

C-reactive protein (CRP) is an acute-phase pentameric reactive protein whose conformation facilitates the ability to trigger complement activation and activate platelets, monocytes, and endothelial cells [81,82]. In humans and animals, CRP, which is a long-established marker of sepsis, is increased in patients with sepsis [83,84,85]. This increase is mainly induced by interleukin (IL)-6 and IL-1β, which increase CRP transcription during the acute phase of the inflammatory process. The low specificity of CRP to sepsis is its main drawback as a biomarker of sepsis in adults, which could explain why this biomarker is not used in veterinary medicine as a biomarker of sepsis to date. In equine medicine, CRP is also identified as an acute-phase protein that has been observed to be increased in arthritis, pneumonia, enteritis, and post-castration [86]. However, since the 1990s, this protein has not been studied in equine sepsis.

#### 6.1.5. Procalcitonin

In vitro and in vivo studies have demonstrated that procalcitonin (PCT), in the context of sepsis, can have harmful effects. Indeed, the presence of PCT in vitro has been shown to have a dose-dependent effect on reducing the phagocytic activity of neutrophils and their ability to migrate to a chemoattractant [87]. In human sepsis, plasma PCT levels increase markedly, and PCT is recognized as a specific and early marker of microbial infection and sepsis [88,89]. Veterinary studies have sought to assess the plasma concentration of PCT in horses. However, data on the role of PCT in equine sepsis are limited. In 2014, Rieger and colleagues developed an ELISA with which they reported significantly higher mean plasma PCT levels in septic horses (*n* = 5) compared with healthy horses (*n* = 24) (8.450 ng/mL versus 47 ng/mL) [90]. In 2015, Bonelli and collaborators investigated PCT levels in sepsis horses and identified that this marker appears to be sensitive and specific to SIRS in horses [91]. However, this study only used 48 horses with SIRS; it would be interesting to reproduce this analysis on a larger population. In addition, this study did not look at sick horses that did not have SIRS, making it difficult to conclude the specificity of PCT. In the same year, Teschner and collaborators confirmed this result by studying naturally endotoxemic horses affected by colic [92]. Similarly, the study of PCT concentrations in plasma and peritoneal fluid of colic horses showed a higher concentration of peritoneal PCT in sick animals compared to healthy animals [93]. However, these studies do not track PCT kinetics, whereas the 2016 Survival Campaign for Sepsis guidelines propose that serial measurement of PCT concentrations can be used to help shorten the duration of antimicrobial therapy in patients with sepsis [94]. Nocera and collaborators assessed PCT levels over time and showed that PCT remained higher in colic horses compared with healthy horses up to 96 h after admission to veterinary teaching hospitals [95]. However, studies evaluating plasma PCT levels in colic horses over a longer period of time would be needed to assess whether this biomarker could be used to guide treatment as in humans.

Acute-phase proteins are used as markers of sepsis in both humans and horses; however, these markers lack specificity (Table 5). Today, there is still no test for early detection of this disease in both human and veterinary medicine. Currently, the diagnosis of equine sepsis is mainly based on the detection of clinical parameters of SIRS without really considering the notion of MODS. However, MODS is now central in the definition of septic shock. The identification of new plasma biomarkers could improve the knowledge of the pathophysiological mechanisms and thus allow the improvement of the diagnosis and the prognosis of sepsis both in horses and humans.

### 6.2. Clinical Perspective: Studying the Secretome to Identify New Biomarkers

Seventy-five years ago, the existence of circulating blood factors promoting depression of myocardial function during septic shock in humans was suggested. In 1985, Parillo and collaborators confirmed this hypothesis by showing that the transfer of serum from patients in septic shock to healthy cardiomyocytes induced a decrease in their shortening rate [96]. It was not until 2000 that the term secretome was first mentioned by Harold Tjalsma and colleagues who were studying the proteins secreted by the bacterium *Bacillus subtilis* [97]. The secretome is a dynamic and complex entity that varies according to the cell type, the functional state of the cell, and time [98]. Indeed, depending on the stimuli they receive, the proteins released by each cell can vary. Thus, the proteins released and their abundance in a given environment can reflect a disease state [99]. Recently, work has shown that the transfer of plasma from patients with septic shock to skeletal muscle tissue results in a loss of myosin, which is not evident with the addition of plasma from healthy patients. There are therefore circulating factors that contribute to this muscle weakness. Pro-inflammatory cytokines such as IL 6, TNF-α, interferon-ϒ, or interleukin-1β are known to be involved in muscle degeneration pathologies. In this study, they highlight a significant negative correlation between myosin content and plasma IL-6. However, the addition of IL-6 alone to control plasma did not induce muscle atrophy [100]. These results, therefore, suggest that other, currently unidentified, circulating factors are involved in the transmission of the pathological phenotype. Thus, the study of the proteins that compose the secretome could lead to a better understanding of the pathophysiological mechanisms, facilitating the identification of new biomarkers in many pathologies such as sepsis. These compounds could be used at an early stage as indicators of pathology development. The plasma secretome is easily accessible by a non-invasive approach and can be analyzed by non-targeted mass spectrometry [11]. Recently, a study identified, through the investigation of the secretome of a rat model in endotoxemic shock, deregulated proteins throughout the course of the pathology [101]. The analysis of the secretome of horses in sepsis could be a source of diagnostic, prognostic biomarkers, and therapeutic targets. Given the complexity of this pathology and the great heterogeneity between equine patients, it would seem more appropriate to identify a protein combination of biomarkers whose levels change during sepsis. Such a combination would improve the sensitivity and specificity of these potential biomarkers. However, before a biomarker’s combination can be used clinically, it must be evaluated through a three-step process as described in the review by Póvoa and Salluh. The first is the analytical validation, which characterizes the laboratory method to be used to measure the biomarker(s). Then the qualification step will allow the evaluation of the evidence that demonstrates an association between the biomarker(s) and the disease. Finally, the last step is the evaluation of the clinical use of biomarker(s) [102].

## 7. Conclusions and Perspective

Animal models have been widely used over the last 20 years to improve our understanding of sepsis and, in particular, to identify biomarkers. However, much of this work has not been transposed to humans, probably because these animal models do not reflect the huge complexity of sepsis in humans. Despite the differences between horses and humans, the spontaneous development of sepsis in horses, especially after gastrointestinal diseases, could be a lever to understand the pathophysiological mechanisms and for the search for biomarkers and therapeutic targets in large-scale studies. It would therefore be interesting to include horses following their hospitalization for sepsis and to study their secretome in order to identify new candidate biomarkers. However, to obtain robust and reproducible results, such studies must rely on two levers. The first lever would be to validate the definition of sepsis used in veterinary medicine by means of an international consensus. As requested in January 2018, this definition should be as close as possible to the definition used in human medicine [103]. In addition, it will also be necessary to standardize the inclusion criteria used in the studies so that they can be compared to each other. The second is the use of large cohorts of horses developing spontaneous sepsis. This perspective requires the mobilization of a maximum of equine veterinary care centers in France and worldwide in order to include a large cohort of horses and to carry out multicenter studies. For example, the Oniris International Horse Health Center in Nantes, France, admits around 120 horses with gastrointestinal disease per year, with nearly half of them showing sepsis signs, so it would take 5 similar equine hospitals to perform a study of 300 cases in 1 year. These conditions will allow the realization of robust and reproducible studies that will allow to better manage this pathology and thus limit the morbi-mortality as well from a human medicine point of view as from a veterinary medicine one.

## Figures and Tables

**Table 1 cells-12-01052-t001:** Assessment of MODS (multiple organ dysfunction syndrome) by the SOFA (sequential organ failure assessment) score adapted from Teggert et al., 2020 [9].

Calculation of the SOFA Score	Score				
	0	1	2	3	4
Respiration PaO_2_/FiO_2_ (kPa)	≥53.3	<53.3	<40	<26.7 with ventilation support	<13.3 with ventilation support
Coagulation Platelets (10^3^/mm^3^)	>150	<150	<100	<50	<20
Hepatic Bilirubin (µmol/L)	<20	20–32	33–101	102–204	≥204
Cardiovascular					
Hypotension (mmHg)	MAP > 70	MAP < 70			
Dopamine (µg/kg/min)			<5	5.1–15	>15
Central Nervous System Glasgow score	15	13–14	10–12	6–9	<6
Kidney					
Creatinine (µmol/L)	<110	110–170	171–299	300–400	>440
Urine (mL/day)	-	-	-	<500	<500

PaO_2_/FiO_2_: partial pressure of oxygen/fraction of inspired oxygen; MAP: mean arterial pressure.

**Table 2 cells-12-01052-t002:** Summary of advantages and disadvantages of induced large mammal models adapted from Guillon et al., 2019 [14].

Models	Advantages	Disadvantages
Rodent	- Less expensive - Wide range of reagents - Wide range of transgenic model - Reduced variability - Short reproduction time	- Resistance to LPS - Zootechnical issues - Anatomy and physiology disparity with human
Canine	- Hemodynamic similarity - Cytokine similarity	- Resistance to LPS - Zootechnical issues - Long reproduction time
Sheep	- Diversity of races - Extremely sensitive to LPS - Cardiopulmonary similarity	- Ruminants - Zootechnical issues - Long reproduction time
Swine	- Omnivorous as humans - Anatomy and physiology similar to humans - Diversity of races - Reproduce the thermogenesis response to stress and the systemic energetic failure associated with septic shock	- Zootechnical issues - Long reproduction time - Pronounced acute pulmonary hypertension

**Table 3 cells-12-01052-t003:** Summary of the SIRS criteria used for adult horses in Schwarz et al., 2012; Aguirre et al., 2013; Epstein et al., 2013; Borde et al., 2014. [35,42,43,44].

	Schwarz et al., 2012	Aguirre et al., 2013	Epstein et al., 2013	Borde et al., 2014
Heart rate (Bpm)	>60			>50
Respiratory rate (Mpm)	>30	>30 or PaCO2 < 32 mmHg	>30	
Temperature (°C)	>38.5	<37.2 or >38.5	>38.6	<36.7 or >38.6
Cell counts of leukocytes (× 109 cells/L)	<5 or >10	<4 or >12.5 or >10% immature GNN	<4.5 or >12.5 or >10% GNN immatures	<5 or >14.5
Additional criteria	/			Lactate > 2 mmol/L or MAP < 90 mmHg

Bpm = beats per minute; Mpm = movements per minute; MAP = mean arterial pressure; GNN = neutrophilic granulocytes.

**Table 4 cells-12-01052-t004:** Overview of threshold values defining SIRS according to Roy et al., 2017 [40].

Clinical Sign	Threshold Values
Temperature (°C)	<37 or >38.5
Heart rate (Bpm)	>52
Respiratory rate (Mpm)	>20 or PaCO_2_ < 32 mmHg
Cell count of leukocytes (×10^9^ cells/L)	<5 or >12.5

Bpm = beats per minute; Mpm = movements per minute.

**Table 5 cells-12-01052-t005:** Summary of advantages and disadvantages of main biomarkers currently used to diagnose sepsis.

Biomarkers	Advantages	Disadvantages
Fibrinogen	- Markers of inflammation - Inexpensive - Easy measurement	- Lack of specificity
Serum amyloid A	- Major acute phase proteins in horses - Distinguish between septic and non-septic inflammatory states - Rapid and pronounced increase in response to inflammatory disease	- Lack of specificity
Procalcitonin	- Increase markedly - Early marker of microbial infection	- Lack of specificity

**Table 6 cells-12-01052-t006:** Summary of comparison of the main biomarkers of sepsis and their changes in humans and horses.

Biomarkers	Human	Horses
Fibrinogen	- 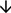 or 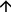 in sepsis [50,51]- 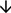 associated with mortality [50]	- 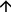 in sepsis- 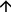 24 h after induction of inflammation
D-dimer	- 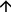 in sepsis	- 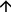 in sepsis
C-reactive Protein	- 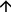 in sepsis	- 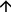 in sepsis
Serum amyloid A	- 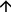 in sepsis [52]	- 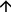 in sepsis- 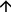 12 h after surgery
Procalcitonin	- 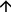 in septic patients- Levels begin to rise 4 h after the onset of systemic infection and peak between 8 and 24 h [53]	- 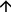 in sepsis- PCT remained higher in colic horses compared with healthy horses up to 96 h after admission

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
