# Peer review of "Spontaneous Sepsis in Adult Horses: From Veterinary to Human Medicine Perspectives"

_cells, 2023, doi:10.3390/cells12071052_

Round 1

Reviewer 1 Report

This manuscript builds on recent studies showing that early diagnosis and prognosis of sepsis is necessary for specific and timely treatment. However, there are currently no predictive biomarkers or therapeutic targets, mainly due to the lack of relevant models. The results obtained in these studies are difficult to transfer to humans due to the pathophysiology of different species. The horse may be a promising model for spontaneous sepsis. This review firstly outlines the different animal species used as models of human sepsis and secondly focuses on adult equine sepsis as a model of spontaneous sepsis and its potential implications for humans and veterinarians.

This work is original and of interest for understanding the sepsis models and for understanding the development of this disease in some cases.

The work investigates which animal model would be suitable for researching spontaneous sepsis.

With this objective, they outline the different animal species and recommended the adult horse model of sepsis.

The discussion is consistent in comparing and discussing the phenomena and possibilities found.

My only observation is that there are 82 articles listed in the paper, is this not enough to elaborate and explain the content of the articles? For some of the studies, it might be better to add some articles.

Author Response

We thank the reviewer for pointing out the lack of references in some section. Equine sepsis remains poorly studied and unlike foal sepsis, studies of interest in adult horse sepsis remain rare, which explains the limited number of references. Some references have been added throughout the manuscript when pertinent, for example as in the paragraph "Main Biomarkers of Equine Sepsis: A Proof of Concept for Human Medicine" to emphasize the comparison between human and equine markers of sepsis.

In order to emphasize the similarity between horses and humans, a paragraph on D-dimer (364-377) and CRP (l 404-415), which are biomarkers regularly used in human clinics, has been added to the paragraph "Main Biomarkers of Equine Sepsis: A Proof of Concept for Human Medicine”.

“6.1.2. D-dimer

Pro-inflammatory cytokines, released during sepsis, activate the coagulation cascade which promotes fibrin degradation resulting in an increase in d-dimer [2]. In human medicine, this protein is used to monitor septic patients particularly to diagnose disseminated intravascular coagulation. It has been demonstrated in humans with sepsis, that this protein is a good predictor of mortality [3]. D-dimer is a simple, inexpensive, commonly available test that reflects closely the activation of the coagulation system and, in this way, potentially also the severity of the host response. In veterinary medicine, D-Dimer has been identified as a useful tool for the detection of thrombotic state and hyper-fibrinolytic states in horses with colic. [4–6]. Furthermore, it has been shown that horses with colic show an increase in peritoneal D-dimer concentration compared to healthy horses [7] and that this protein could be used as a diagnostic and prognostic marker for horses with severe forms of colic [8]. However, to date, D-dimer is rarely used as a biomarker of sepsis in the adult horse.”

“6.1.4. C-reactive protein

C-reactive protein (CRP) is an acute-phase pentameric reactive protein whose conformation facilitates the ability to trigger complement activation and activate platelets, monocytes and endothelial cells [12,13]. In humans and animals, CRP, which is a long-established marker of sepsis, is increased in patients with sepsis [14–16]. This increase is mainly induced by interleukin (IL)-6 and IL-1β which increase CRP transcription during the acute phase of the inflammatory process. The low specificity of CRP to sepsis is its main drawback as a biomarker of sepsis in adults, which could explain why this biomarker is not used in veterinary medicine as a biomarker of sepsis to date. In equine medicine, CRP is also identified as an acute phase protein that has been observed to be increased in arthritis, pneumonia, enteritis and post castration [17]. However, since the 1990s this protein has not been studied in equine sepsis.”

Reviewer 2 Report

Comments to the Authors of manuscript number: cells-2211603 entitled “Spontaneous sepsis in adult horses: from veterinary to human medicine perspectives”.

Reading the abstract and following the title there is a moment when someone has to stop and try to understand deeper. It is very good idea presented by the review. Moreover, the sepsis in horses is difficult issue. Authors have presented also other animal models of sepsis, their disadvantages and advantages were discussed as well as similarities to human sepsis. It is worth to add some information about fibrin D-dimer in the part of fibrinogen and its use in the monitoring of septic patients, and whether it is used in diagnosis of sepsis in horses.

The whole text is written in good manner, it is easy to understand and tracking the train of thought.

It is very interesting description, worth to publish.

Author Response

We thank the reviewer for this comment. Studies have proposed D-dimer as a biomarker for equine sepsis, but to date it is not used as such in veterinary medicine. To clarify this point, a paragraph has been added after the one on fibrinogen (l 364-377).

“6.1.2. D-dimer

Pro-inflammatory cytokines, released during sepsis, activate the coagulation cascade which promotes fibrin degradation resulting in an increase in d-dimer [2]. In human medicine, this protein is used to monitor septic patients particularly to diagnose disseminated intravascular coagulation. It has been demonstrated in humans with sepsis, that this protein is a good predictor of mortality [3]. D-dimer is a simple, inexpensive, commonly available test that reflects closely the activation of the coagulation system and, in this way, potentially also the severity of the host response. In veterinary medicine, D-Dimer has been identified as a useful tool for the detection of thrombotic state and hyper-fibrinolytic states in horses with colic. [4–6]. Furthermore, it has been shown that horses with colic show an increase in peritoneal D-dimer concentration compared to healthy horses [7] and that this protein could be used as a diagnostic and prognostic marker for horses with severe forms of colic [8]. However, to date, D-dimer is rarely used as a biomarker of sepsis in the adult horse.”

Reviewer 3 Report

The review paper Spontaneous Sepsis in Adult Horses: From Veterinary to Human Medicine Perspectives reviewed animal models widely used over the last two decades to improve our understanding of sepsis and in particular to identify biomarkers. Authors are pointing out that although big volume of work has been done in this area, the results have not been successfully translated to humans, because these animal models do not reflect complexity of sepsis in humans. The spontaneous development of sepsis in horses, especially after gastro-intestinal diseases, could serve as appropriate model and considerably increase our knowledge of the mechanisms and biomarkers that could be used as therapeutic targets in sepsis.

Comments:

1.      The paper is interesting and well structured. Use of equine sepsis as a proof of concept for human medicine represents valid and interesting approach.

2.      Main biomarkers of equine sepsis are listed, and summary of their advantages and disadvantages discussed and presented in the Table V. It will be interesting to summarize comparison of main biomarkers and their changes in humans and horses, in Table format.

3.      Role of specific biomarkers listed in the review in sepsis should be expanded to include mechanisms of their role in sepsis. Role of pro-inflammatory cytokines in sepsis should also be discussed in more details.

4.      Illustration depicting similarities (and dissimilarities) of sepsis parameters in human and equine sepsis would be great visual addition to the review conclusions.
